# The Communicative Effectiveness of Branding at Sports Press Conferences

**DOI:** 10.3390/bs12050151

**Published:** 2022-05-18

**Authors:** Galvão Meirinhos, Tiago Mendes, Reiville Rêgo, Márcio Oliveira, Margarida Rodrigues, Rui Silva

**Affiliations:** 1LABCOM-IFP, University of Trás-os-Montes and Alto Douro—UTAD, 5000-801 Vila Real, Portugal; galvaomeirinhos@gmail.com (G.M.); tiagomendes47@outlook.com (T.M.); 2Universidade Federal Rural da Amazônia/UFRA–Campus Tomé-Açu, Tomé-Açu-PA 68680-000, Brazil; reiville.rego@ufra.edu.br; 3NECE—Research Unit in Business Sciences, ISLA Santarém—Instituto Superior de Gestão e Administração de Santarém, Largo Cândido dos Reis, Santarém 2000-241, Brazil; marcio.oliveira@ipleiria.pt; 4Polytechnic Institute of Leiria, Rua General Norton de Matos, Apartado 4133, 2411-901 Leiria, Portugal; 5CEFAGE-UBI Research Center, University of Beira Interior, Estrada do Sineiro, 6200-209 Covilhã, Portugal; mmmrodrigues@sapo.pt; 6Department of Economy, Sociology and Management, University of Trás-os-Montes and Alto Douro, 5001-801 Vila Real, Portugal

**Keywords:** branding, sport press conferences, communication

## Abstract

This scientific work studies brand placement in the press conferences of soccer coaches and evaluates their communicative effectiveness through the measurement of their cognitive and affective effects on the viewers. In this research, we established the following objectives: (1) to examine the characteristics of the practice of brand placement in football press conferences: the diffusion times of brands, space occupied on the screen, and categories of brands placed; (2) to evaluate the behaviour of the human eye when viewing press conferences, in terms of continuous movements (saccades) and fixations (fixations) on brands; (3) to gauge the spontaneous and assisted recall of brands by subjects; (4) to verify the correlation between the persistence of visual fixations and recall/recognition; (5) to investigate the changes in subjects’ attitudes towards brands viewed in the experimental context. An exploratory observation was made that enabled a more in-depth knowledge and implementation of brand placement at sports conferences. For the experimental observation, a 2 × 2 factorial design of independent groups with total randomization was defined in order to perceive the influence of the variables “time” and “quantity” on the communicative effectiveness of the placed tags. In order to collect the data, a combination of several tested and validated tools was used, namely the screen division grid in surface units, as advocated by Bravo (1995); the technology of eye-tracking as an instrument for the recognition of the ocular movements of subjects in the observation space; surveys tested for cognitive gauging; and a semantic differential scale to assess attitudes toward the brand. The results indicate that the subjects recall in a spontaneous and suggested way the brands placed at the press conferences and develop positive attitudes about them. The recall is influenced by the diffusion time of the stimulus, and above all, the type of placement on the screen is decisive. It was not found that the brands to which subjects develop more positive attitudes were the most remembered. Finally, the face of the soccer coach is the main focus of attention of the subjects, and the areas surrounding this interlocutor are the ones that arouse the most interest in terms of the placement of brands.

## 1. Introduction

Brand placement is the paid introduction of products or brands in mass media programming [1]. In its most natural form, the placement is integrated into the action, and its presence becomes logical and even indispensable [2]. This is a promotion strategy that has been increasingly used in Portugal. Advertisers have started to invest in new media (online, events, mobile marketing) and in product/brand placement. In the United States of America, investment in product placement in the media grew from 190 million dollars in 1974 to 512 million dollars in 1985 and to 1.13 billion dollars in 1994, to reach 3.458 billion dollars in 2004 [2]. In the United Kingdom, product placement was only allowed from 2011 and already had investments in the order of 24 million pounds, a figure that is expected to increase to 120 million pounds in five years, which is equivalent to 3% of the value of the television advertising market [3]. As we can see, this rise is not only in the United States of America but all over the world, which results from the constant emergence of brands and products and in the willingness of advertisers to channel advertising investments to new contexts and means of dissemination. In brand placement where the product or brand is inserted in a context and television or film plot without normally assuming a verbal or visual protagonism the attempt at persuasion is not explicit [4]. Normally, consumers recognize the persuasive intent of advertising and consequently create scepticism and counter-arguments against its assimilation. On the contrary, in placement, this resistance does not exist as its commercial intent is not easily identifiable—which defines its effectiveness [5]. In view of this, it is fair to say that product or brand placement should not be used as a tool to increase sales but rather as “a tool (...) to create and strengthen brand value” [6].

Some authors indicate four moments when companies use product placement: to change market positioning, improve their image, increase or maintain awareness, and launch new products [7]. In this list, we verify the absence of the use of placement as a tool to increase sales. After all, the placement does not necessarily have to be seen as a substitute for traditional advertising or other sales stimulation mechanisms. In fact, it is increasingly becoming an integral element of the brands’ communication mix [5]. This growing importance is reflected in the greater control that companies wish to have over the placement of their products and brands, so the current practice is already far from the initial concept of mere inclusion for convenience in a film or television production [8].

Football press conferences are times and spaces par excellence for brands to contact their audiences. Through the media and with journalistic treatment, brands communicate and express their support for a club or a sports competition, in most cases through the presence of their minimum communication unit: the logo.

In 2000, Avery and Ferraro [9] carried out a study on the number of brands that appeared on television during prime time. They concluded that sports programmes, despite representing only 1.3% of broadcasting time, accounted for 7.9% of branding, with an average of 58.25 brands every half hour. In addition, information programmes also obtained a significant percentage of branding—20.6% in 8.4% of broadcasting time, adding up to 36 brands every half hour [9]. Although many of these insertions have no direct relationship with the product or brand placement phenomenon, they highlight the importance of sporting and news events as marketing communication vehicles. Scientific research on brand placement is still recent, and there are multiple approaches and different tools that allow the evaluation of communicative effectiveness. The aim of the researcher may thus be to check the influence of the presence of brands in the psychological processes of learning, feeling, or doing [10], through variables such as recall, recognition, and the effect on attitudes or behavioural changes [11]. To this end, it has at its disposal tools that assess visibility, perception, points of attention, and impact on sales. Within the spectrum of possible measures, explicit measures are the most suitable for tests aimed at anticipating behaviour motivated by conscious decisions. Implicit measures are useful for predicting spontaneous behaviour [12]. Implicit measures assume particular importance at the level of conditioning attitudes and purchase behaviour since in most cases, processing of the placement occurs unconsciously [13,14]. The level of recall of a brand/product does not necessarily reflect its effectiveness on an affective and behavioural level [15]. In fact, it may even have the opposite effect, including the generation of negative attitudes [16,17,18].

Our approach involves combining different psychophysical variables in order to test the interaction between these two realities and their influence on the communicative effectiveness of the placement. In this article, we refer to this phenomenon as brand placement since we agree with [1] when he states that product placement, although referring to a product placed in a film, is actually a brand placed in a film. After all, companies are usually more interested in promoting their brands than in a particular product. He also defines brand placement as the paid introduction of products or brands in mass media programming [1]. In its most natural form, the placement is integrated into the action, where the brand’s presence becomes logical and even indispensable [2]. Our own interpretation of the phenomenon leads us to define brand placement as a brand communication tool that consists of the planned and paid placement of indicators of a brand in media productions or events in order to obtain a specific auditory or visual impression or both exposure and intensity.

## 2. Methodology

### 2.1. Research Problem and Hypotheses

Just and Carpenter (1976) [19] argue that what the human eye fixates on is what is being processed by the operational memory, and the longer the fixation time, the longer the processing time. Gupta and Lord (1998) [20] present prominence by size or positioning on the screen as a characteristic of prominent placement, as opposed to a subtle one, which appears in the background, peripheral to the action or with little size and exposure time (Blondé and Roozen, 2007) [21]. Thus, in view of these principles, we pose the research problem as follows:

Does the communicative effectiveness of brand placement in sports press conferences vary as a function of exposure time and type of brand placement?

In order to answer the research problem, it becomes necessary to build several hypotheses that, by their specificity, can contribute to a general perception of the phenomenon under observation. As such, we propose the following general hypothesis:

Brand placement generates cognitive and effective effects on the viewer.

In order to be rigorously proved, this general hypothesis was broken down into operational hypotheses:

**H1.** 
*Brand recall varies proportionally according to the spatial and temporal presence in football press conferences;*


**H2.** 
*Brand recall varies proportionally according to fixations;*


**H3.** 
*Brand recall varies proportionally according to attitudes towards brands.*


In view of testing the operational hypotheses, the observation was oriented in order to prove the existence of different sets of assumptions. Thus, for Hypothesis 1, it was necessary to ascertain whether:(a)Brand recall is greater the longer the exposure time;(b)Brand recall is greater the larger the area occupied by the logo;(c)Brand recall is greater the closer the logo is to the interlocutor.

To test Hypothesis 2, it was verified whether:(a)The human eye performs more fixations on brands when they have a large dimension;(b)The human eye fixates more on brands that are closer to the interlocutor;(c)Brand recall is higher as the number of fixations increases.

To test Hypothesis 3, it was necessary to understand whether:(a)Attitudes vary after being subjected to the stimulus, compared to the exploratory observation tests;(b)The brands towards which subjects develop more positive attitudes are more remembered;(c)The brands with more top-of-mind recall are the ones that subjects know best.

In order to achieve these objectives and obtain answers to the problem by testing the hypotheses put forward, we designed an experiment based on a 2 × 2 factorial design. For this study, four experimental groups were created resulting from the combination of two levels of two independent variables (2 variables × 2 levels). It should also be noted that this is a 2 × 2 factorial design of independent groups and total randomisation, formulated with the purpose of assessing only the primary effects that the manipulation of the independent variables has on the dependent variables while ensuring an equal probability that each of the individuals may be indiscriminately included in each of the independent groups and that each of them may be assigned any of the experimental treatments [22].

#### 2.1.1. Sample

When selecting individuals for the sample, in order to optimise the observation process by reducing it in time, the sampling technique used in this study was accidental sampling, which “consists of subjects or elements that are easily accessible and present at a specific moment” [23]. In this case, these are students from the University of Trás-os-Montes e Alto Douro who were chosen because they were present on the university campus during the observation period and near the site of the experiment.

Thirty-two participants were chosen and randomly distributed into the four experimental groups. The determination of the total of thirty-two participants arises from the idea instituted within the scope of eye-tracking studies that five users are enough to “detect approximately 85% of the problems in an interface, given that the probability of a user finding a problem is about 31%” [24]. Furthermore, the company “Think Eyetracking” states that thirty users are enough since there were no significant differences in previous experiments in the results obtained with a sample of 150 participants and the results of 4 groups of 30 [24].

#### 2.1.2. Communicational Stimuli

For the construction of the communicational stimuli, we appointed as variables to manipulate (independent) the exposure time to the placement of brands (or of viewing the video of a simulated press conference), identified as “time,” and the number of logos present on the screen, identified as “quantity.” Thus, and according to the data collected in a previous exploratory observation, we determined two levels for the “quantity” variable: between 12 and 25 logos and between 25 and 50 logos. The first level consisted of 25 logos in the first half of the stimulus and 12 in the second half. The second level presented 50 logos in the first half of the stimulus and 25 in the second half. For the “time” variable, we established two brand diffusion durations: approximately one minute (59 s) and approximately two minutes (2 min and 13 s).

The construction of the communicational stimuli followed first the research objectives, then the literature review, and, finally, the exploratory observation. If the main objective is to understand the communicative effectiveness of brand placement in football coaches’ press conferences, the literature review tells us that the exposure time and the type of brand placement have an influence on recall and recognition. As evidenced by Ansons (2010) [25], the sharper, larger, and more central, prolonged, repeated and integrated into the scene, the more effective the placement is at the level of memory, as factors that enhance brand perception end up producing positive effects on familiarity and recall. The prominence and length of the placement of the brand influence recall or recognition in a direct proportionality relationship, i.e., the larger and more prolonged, the greater the effect on memory. On the other hand, the exploratory observation provided us with enough data to build four prototypes of the press conference in which real commercial brands that are part of the Portuguese daily life are presented. More specifically, through exploratory observation, we realized that football press conferences are broadcast in the reportage of generalist television news programs in Portugal with a set of characteristics that can be catalogued, such as the type of plan, the exposure time of brands, or the number of brands present in the screen. 

Overall, the images of press conferences are captured based on two types of framing: medium shot and detailed shot. In the exploratory observation, we found that out of a total of 119 selected shots of press conferences, 64 were detailed shots, and 53 were medium shots. Only two of the 119 conference plans analysed contained close-ups. We also found that there is an average of 25 placements per plan (11 showing the logo in full and 14 in part), with the minimum being six and the maximum 54. As we mentioned before, these values vary according to the type of framing: the closer to the interlocutor, the more logos are visualized. As for the brands’ diffusion time, in the exploratory study, we verified that the average diffusion duration of the brands in each news item was around 45 s, and each news item had an average duration of 1 m 52 s. Therefore, the brands present in football press conferences are broadcast for about 40% of the time of the news prepared from the declarations of football coaches in the press room.

Regarding the quantity of logos placed in the football press conference prototypes, the average and intermediate number ascertained in the exploratory observations was 25, and we set a maximum ceiling of 50 placements and a minimum of 12, i.e., double and approximately half of 25. 

For the experiments with a maximum of 50 logos, all the real commercial brands previously involved in the test of the attitudes of the exploratory observation were selected. They were placed, obeying an equitable distribution criterion according to the global average level of the attitudes verified in those tests. Thus, the brands were divided into groups of five, according to the classification obtained in those tests. Following a horizontal distribution sequence in the grid, one mark from each group was placed alternately. This way, we ensured that all marks had the same probability of being outside the screen, behind the couch or in a completely visible place (see Figure 1).

With this random distribution, it is also possible to understand whether the location on the screen has more influence on eye movements, recall, and recognition than familiarity or a positive attitude towards brands. In other words, we can verify whether it is the case that the more subjects look at, remember, or recognize a brand the better their attitude towards it, or if it is enough that it is placed in a privileged position on the screen. Since, as we have already mentioned, the type of framing varies the number of logos presented in football coaches’ press conferences, for each communicational stimulus, we also varied the framing according to the number of logos we wanted to present. Thus, in each experimental group, the stimuli were presented first in a medium plane and then in a detailed plane. This relationship of proportionality existing in the stimuli between the number of marks and the size of the logos on the screen is shown by placing the grid proposed by Ángel Rodriguez Bravo (1995) [26] in Figure 2, Figure 3 and Figure 4. The closer the frame gets to the interlocutor, the fewer brands are visualized and the bigger the size of the logos.

The football coach is an experienced lecturer from the University of Trás-os-Montes e Alto Douro, who is actually also a football coach of a professional football team. He was selected because he combines both situations, understands the reality of scientific studies, as well as being used to the characteristic tasks of real press conferences.

#### 2.1.3. Data Collection Tools

As one of the goals of this research is to verify the existence of visual patterns that clearly influence the recall and recognition, more or less consciousness, of brands, the experimental observation of this study was carried out through a typical eye-tracking study, in which three data collection tools were used: an eye tracker system, a questionnaire, and a semantic differential scale. 

The eye tracker was essential to verify patterns of focus location and time, trajectories or preferences for certain areas of interest (AOI), which are the most relevant aspects for the type of experiment we proposed to perform. With this tool, it was possible to accurately perceive when subjects set the brands. The second data collection tool used in this study was the questionnaire survey. The questions involved the indication of spontaneous and assisted recall of the brands present in the visualized conference. Finally, a semantic differential scale was used, which presented sets of opposite adjectives to which subjects had to indicate how they identified with the brands presented in the communicational stimulus.

## 3. Analysis and Discussion of Results

Because sample normality was not verified, non-parametric tests were performed, as they do not require the distribution of the variable under study to be known and are maybe more powerful for samples of small and different sizes.

### 3.1. The type of Brand Placement as a Factor of Communicative Efficacy

Hypothesis 1 questioned whether brand recall varies proportionally according to spatial and temporal presence in football press conferences. As mentioned in Section 3.3, this hypothesis was subdivided into operational hypotheses. 


**Time**


The first operational hypothesis led us to check whether the brand recall is higher the longer the exposure time. To test this hypothesis, we conducted a Mann–Whitney test to see if the medians of the groups were significantly different for the variables “spontaneous recall” and “assisted recall”.

In Table 1 of the classifications, it can be seen that for the independent variable “time” all the dependent variables obtained higher averages in the two-minute groups.

The *p*-values are greater than α = 0.05 in both dependent variables: 0.669 in spontaneous recall and 0.160 in the assisted recall. Therefore, the null hypothesis that the distribution of the dependent variables is the same in all categories of the variable “time” is retained. The exact *p*-value was considered since the sample size is small (Table 2).

Once the absolute values of recall were evaluated, we proceeded to the analysis of the recall values for each brand. We then measured the total exposure time for each brand, checking the number of placements of the brands in each of the two parts of the experiment’s communicational stimuli. Thus, we checked the placements, regardless of their size, for every 34 and 25 s, in the approximately 1-min videos and for every 64 and 68 s in the approximately 2-min videos.

Establishing the correlations between time (total_segs) and spontaneous recall (SR) and assisted recall (AR), in the table above, it can be seen that the exposure time of brands establishes strong correlations with spontaneous (0.702) and assisted (0.687) recall. This correlation indicates that brand recall is as high as the exposure time (Table 3).


**Quantity**


The second operational hypothesis posed the question of whether the brand recall is greater the more area occupied by the logo. In order to check this assumption, we performed a Mann–Whitney test to see if the medians of the groups were significantly different for the variables “spontaneous recall” and “assisted recall”.

With the variable “quantity” there were differences between the groups in all dependent variables. The groups with 25–50 logos obtained higher averages for spontaneous recall and assisted recall (Table 4).

Spearman correlations were performed to check whether the values for spontaneous recall and assisted recall are associated with the quantity variable. Statistically significant correlations were found to exist, given that the *p* values were below α = 0.05. There are, then, moderate correlations between the quantity of logos and spontaneous recall (0.411) and assisted recall (0.388) (Table 5).

After checking the absolute recall values, we moved on to the recall values per brand. We then evaluated the space occupied in surface units, according to Bravo’s model (1995) [26], in the totality of the four communicational stimuli of the experiment.

Establishing the correlations between the space occupied in surface units (total_US) and spontaneous recall (SR) and assisted recall (AR), in the table above, it is verified that the space occupied by brands establishes strong correlations with spontaneous (0.724) and assisted (0.709) recall. This result is in line with the operational hypothesis, which states that brand recall is higher the more space is occupied on the screen (Table 6).


**Position**


The third operative hypothesis questioned whether the brand recall is greater the closer the proximity to the interlocutor.

In order perceive how the position of the brands is related to recall and fixations, we performed Spearman correlations. We concluded that there were strong correlations between the score that classifies the position of the brands (US_Pont_Tot), spontaneous recall (0.747), and assisted recall (0.732) (Table 7).

### 3.2. Fixations as Predictors of Brand Recall

Hypothesis 2 asked whether “brand recall varies proportionally according to fixations.” To gauge this assumption, we sought to answer three operational hypotheses.


**Dimension**


The first operational hypothesis guided us to understand if the human eye performs more fixations on brands when they have a larger dimension. To test this hypothesis, we performed the Mann–Whitney test, observing whether the medians of fixations of the groups were significantly different in the variables according to the number of logos (Table 8).

Differences in average fixation ranks are observed between the groups who viewed videos with 12 or 25 logos (larger size) and those who viewed with 25 or 50 logos (smaller size). The latter obtained higher average fixations than the former (Table 9).

The differences in the values obtained are statistically significant since the *p*-values are less than α = 0.05. For fixations, the *p*-value = 0.012, which leads us to reject the null hypothesis that the distribution of the dependent variables is the same for both categories of the variable “quantity” (Table 10).

Spearman’s rho values indicate that there is a moderate correlation, of 0.447, between the amount of logos and fixations, statistically significant, as *p*-values are less than α = 0.05.

Establishing the correlations between the space occupied in surface units (total_US) and fixations (total_fixs) in the table above, it can be seen that there is a strong, statistically significant correlation (0.790) between both variables (Table 11).

Position

The second operational hypothesis called into question whether the human eye performs more fixations on brands that are closer to the interlocutor.

From the table it can be seen that there is a strong positive correlation (0.845) between the score (US_Pont_Tot) obtained by the brands, according to their proximity to the interlocutor, and the number of fixations (Fix-tot) on each of them (Table 12).


**Recall**


The third operational hypothesis aimed at the perceived relationship between brand recall and the number of fixations.

There are moderate correlations between the total number of fixations (Fix_tot) and spontaneous recall (SR), of 0.572, and also with assisted recall (AR), of 0.635) (Table 13).

Of the statistically significant correlations (*p* ≥ α = 0.05) we highlight: the moderate positive correlation of the independent variable “quantity” with fixations (0.447), spontaneous recall (0.411) and assisted recall (0.388); the moderate correlation of fixations with spontaneous recall (0.547) and strong with assisted recall (0.690) and a strong correlation between spontaneous and assisted recall (0.832). The variable “time” does not correlate in a statistically significant way with any dependent variable.

### 3.3. Attitudes as a Factor in the Recall

Hypothesis 3 guided the study in the search for the perception of the variation in brand recall according to attitudes towards them. This hypothesis was subdivided into operational hypotheses.


**Variation**


The first operational hypothesis stated: “attitudes vary after being subjected to the stimulus, compared to the exploratory observation tests.” To verify this hypothesis, we carried out an analysis of the variations in attitudes. According to the semantic differential scale, we presented the subjects who participated in the exploratory observation and those who participated in the experimental observation. 

We verified that the global averages of most of the brands obtained higher values in the experimental observation. The overall average was 4.2% higher in the experimental observation.

As the table shows, the mean values of attitudes are higher in the experimental observation (125.23), compared to the exploratory observation (97.03). In order to confirm these differences, we performed the Mann–Whitney U-test (Table 14).

Taking into consideration a grouping of the values according to the phase of the study in which they were collected (exploratory or experimental), it can be seen that the medians of the groups that resulted were significantly different, as the *p*-value is less than α = 0.05 (0.012) (Table 15).


**Recall**


The second operational hypothesis asked whether “the brands towards which subjects develop more positive attitudes are remembered more.” We obtained the results necessary to confirm this hypothesis by correlating the mean values of subjects’ attitudes towards brands with those of the spontaneous and assisted recall of the same. The brands with the best average attitudes are not the most remembered ones (Table 16).

We conclude that there are significant moderate correlations between the mean value of attitudes towards brands and spontaneous recall (0.676) and assisted recall (0.688).


**Top-of-Mind**


The last operational hypothesis aimed to assess if “the brands with the highest top-of-mind recall are the ones most known by the subjects.” We performed correlations between the top-of-mind values of the brands and the average values of all subjects in the attitudes of knowledge and familiarity towards them. We observed that brands that obtained average values of knowledge higher than 4.50 were not remembered in a top-of-mind (TOM) fashion by the subjects.

Having established the correlations, it can be seen that there is a moderate (0.414) and significant (*p* = 0.044 < α = 0.05) correlation between brand knowledge and the fact that they are recognised first by the subjects spontaneously (top-of-mind) (Table 17).

## 4. Discussion

What is verified in the first instance is that the placement of the brands has a cognitive effect on the spectators since the subjects of the experiment indicated significantly that they spontaneously remembered them and, a posteriori, recognized them through suggestion. These are conclusions that are not surprising based on previous studies but that bring new information to this new reality under study: that of football coaches’ press conferences.

With Hypothesis 1, “brand recall varies proportionally according to the spatial and temporal presence in football press conferences,” we found that there is a correlation between the way brands are placed and the cognitive effect, especially in terms of recall, that they have on viewers. This conclusion especially was checked through the methodological proposal presented for quantifying the prominence of the placement. We noticed then, through the comparison between the experimental groups, that (a) the exposure time is not determinant for a higher number of brands to be remembered since no correlation was found between the duration of the stimulus and the number of brands remembered. On the other hand, through internal analysis, comparing the results of recall of the placed brands, we found that (b) it correlates with exposure time because the recall of a brand varies according to the exposure time. The longer the exposure time, the higher the levels of brand recall. With regard to the number of brands present on the screen, it can be seen that (c) in the intergroup analysis, the larger number of logos present leads to the recall of a larger number of brands because groups with a larger number of logos have higher recall rates. On the other hand, this information shows that a larger occupation of the screen by brands does not translate into a larger amount of recall. On the other hand, in the analysis between the placed brands, we conclude that those that occupied more screen space (measured in surface units) were the ones with higher recall rates. The same happened with the positioning of the brands, indicating that those which are located closer to the speaker are the ones that have a higher probability of being remembered.

The results obtained for Hypothesis 1 are in line with the conclusions reached by Ansons (2010) [25], which indicate that the clearer, larger, more central, longer, repeated, and integrated into the scene the placement is, the greater its effectiveness in terms of memory because the factors that improve the perception of brands end up producing positive effects on their familiarity and recall. This is also the idea postulated by Lehu (2007) [2], who refers to space, time, and the number of occurrences as determining factors for the prominence of the placement of brands and, consequently, for the spectators’ recall. Thus, it is verified through this study and confirming Hypothesis 1 that brand recall varies proportionally according to the spatial and temporal presence in football press conferences.

For Hypothesis 2, “brand recall varies proportionally according to fixations,” it was found that (a) the groups subject to viewing more logos are the ones fixate more often. On the other hand, (b) the brands that occupy more space on the screen are those that elicit a greater number of fixations. Likewise, (c) the positioning of the logo closer to the interlocutor results in more fixations on the brands. In general, there was a moderate positive correlation between fixations on brands and brand recall since the most fixated brands tend to be the most recalled. This confirms the hypothesis that brand recall varies proportionally according to fixations since the prominence of brand placement, as verified with Hypothesis 1, leads to a higher level of recall. Furthermore, according to the test results for Hypothesis 2, the most prominent locations, i.e., those that occupy more space and are closer to the interlocutor’s face, are those that result in a higher number of fixations, as we had the opportunity to verify through the data collected with the eye-tracker.

Regarding Hypothesis 3, “brand recall varies proportionally according to attitudes towards brands,” we found that attitudinal values, in general, were higher in the experimental observation after being subjected to the stimulus, compared to those that resulted from the exploratory observation survey. It was found that the most remembered brands are not necessarily the ones towards which subjects develop more positive attitudes. Regarding this aspect, there was a moderate positive correlation between recall and attitudes. Furthermore, the most recalled top-of-mind brands have a low correspondence with the ones subjects are more familiar with, and there is a moderate correlation between top-of-mind recall and brand awareness.

These results for Hypothesis 3 confirmation are in line with the literature analysed, which indicates a dissociation between brand recall and attitudes towards them. Van Reijmersdal (2009) [27], for example, concluded precisely this by finding that the effects that exposure to brands had on attitudes towards them had nothing to do with memory. Attitudes were affected without there being a recollection of viewing them. The development of attitudes was a result of the mere-exposure effect, in which, according to Zajonc (1968) [28], repeated exposure to a stimulus is a sufficient condition for there to be an improvement in attitude.

In summary, the main results of this study reveal that the communicative effectiveness of brand placement in sports press conferences is greater the more space each brand occupies on the screen and the longer the exposure time. This is because it turns out that a larger size, longer exposure, and greater proximity to the main object (in this case, the football coach) lead to more fixations and, consequently, conscious processing of the placed brands, which results in higher levels of recall. This recall happens with a low level of dependence on attitudes towards the brand since familiarity with them does not always result in higher levels of recall.

## 5. Conclusions

In this study of elaborate methodological complexity, within the scope of the social sciences, namely communication, it was possible to reach a considerable set of conclusions.

The analysis methodology used allowed us to break into unexplored territory in the field of communication sciences. The application of the experimental method, using data collection tools like the eye-tracker to study the communicative effectiveness of brand placement in football coaches’ press conferences, constitutes a new context. Besides psychological data associated with memory and attitudes, usually collected with questionnaire surveys, we sought to obtain records about physical activity relevant to the object of study. Considering that vision is fundamental for the recall of brands placed on the screen, we used a non-invasive eye tracker to monitor eye movements. This allowed us to assess the agreement, or not, between vision, memory, and attitudes towards the brands placed on the screen. With this set of tools, we also adapted the Cuadro–Pantalla method in order to divide the screen into one hundred surface units, which allowed us to check where the subjects directed their attention most of the time through the fixations and saccadic eye movements detected by the eye tracker. Thus, we noticed that the most interesting place for placing brands is near the interlocutor’s head, which is where subjects direct their gaze most of the time. The hypotheses of the study were verified by means of an experiment with a 2 × 2 factorial design of independent groups with total randomization. Currently, brands pay for their introduction in films and television programs according to the size of the audience and the expected return based on that same number. What we propose with this study is the elaboration of a methodology that allows the evaluation of their communicative effectiveness in a more objective way with regard to memory and the elaboration of attitudes.

It was found that brand recall varies proportionally according to the spatial and temporal presence in football press conferences, as there is a correlation between the way brands are placed and the cognitive effect, especially in terms of recall. It was noticeable through the comparison between the experimental groups that the exposure time to the brands did not cause a higher number of (diverse) occasions of recall. However, the brands with more exposure were more often recalled. This suggests that there is an effect of time on memory. On the other hand, space is also important for brand recall since the brands with more space were remembered more. The same happened with the positioning of the brands, indicating that those located closest to the speaker were the ones most likely to be remembered. This positioning and size had an equal influence on the number of fixations by subjects. The more central and larger, the better. It means that the most consciously seen and processed brands are the most remembered.

As far as attitudes are concerned, however, we found that the most remembered brands are not necessarily the ones towards which subjects develop more positive attitudes. Furthermore, the top-of-mind recalled brands have a low correspondence with those subjects they are more familiar with. This indicates that brand placement has an effect on eye movement and recall, but its influence on the development of attitudes towards brands is minimal. This is especially relevant if we consider that the brands towards which subjects have positive attitudes are not the most remembered.

We can thus state, in response to our research question, that brands placed in football press conferences provoke cognitive effects on viewers. On the other hand, they trigger effective effects but with little relevance.

Press conferences are, both in an encrypted and open channel, a focus of great interest, given their repeated and extensive use by television channels. Going back to the previous example, about 70% of the Spanish population watches the news on television, a market too vast to be wasted by advertisers. Thus, it becomes practically impossible to have press conferences where brands are not present. Behind the interviewee or in front of him, on a fixed panel or digital, there are several ways to try to capture the viewers’ attention to the logo or the presence of the product. Among the many reasons for this research, the benefits of knowing better the communicative and advertising capacities of press conferences are immense, especially for advertisers, advertising agents, and sports organisations. Only by assessing, qualitatively as well as quantitatively, the potential value of press conferences related to sporting events can we know the real value in terms of communication and in terms of advertising exploitation.

## 6. Limitations and Future Research

When conducting a research study like this, which is both comprehensive and in-depth, we faced some difficulties and decisions that necessarily limited the boundaries to which we had to adjust the development of our work. Thus, of the main limitations, the first came from the lack of studies with a similar analysis methodology, which made comparison and validation difficult. The second was related to the limited availability of volunteers to participate in the study, which caused the work to be prolonged. The last and most relevant limitation arose from the scope of the variables that could have been studied and which, due to the size of the research corpus, we had to opt not to verify, such as purchasing behaviour, for example.

The results achieved in this research and the conclusions that we reached enable us to launch some recommendations regarding both the implementation of brand placement at sports press conferences and the development of new studies. Regarding the implementation of brand placement in sports press conferences, the conclusions indicate that brands should be as large as possible and placed as close as possible to the main focus of the spectator’s attention, which, in the case of sports press conferences, is the coach. They should also be placed in contexts that guarantee plenty of exposure time. We verified, then, that the more brands are placed, the greater is the dispersion of the gaze, so it is favourable for brands that there are few logos on the screen so that those placed appear with large dimensions and closer to the interlocutor because these are the ones that get higher rates of fixation and recall by the spectators. Therefore, brands should try to exert influence so that the preparation of panels or devices for displaying logos takes into account these factors of communicative effectiveness of brand placement, which are also influenced by the placement of television cameras and the positioning of the trainer.

Exposure time is another determining factor for communicative effectiveness. However, the time on television is short. Furthermore, there is no control over the airtime of brands in press conference news since this is a decision made by the journalist when editing the news. As a result, neither brands nor press conference promoters can control the length of exposure. Alternatively, brands should seek agreements with clubs and institutions that organize the competitions in order to try to obtain larger dimensions for their logo and its proximity to the speaker, as indicated above.

We also found that attitudes do not influence the level of recall of brands placed in sports press conferences. This fact puts all brands on an equal footing, whether they have a high or low level of awareness. This shows that brand placement in sports press conferences can be a very useful tool for brands that are entering the market or want to maintain their image with their target audience.

Regarding recommendations for future research, we believe that it would be interesting if the analysis methodology applied in this research was replicated with a larger and more diverse sample in order to try to find a representative sample of the population under study. Other variables that may influence the communicative effectiveness, such as chromatic variations, the type of placement (in the panel or in digital format) or the location (behind, in front, or on the coach’s clothing) may also be considered. Finally, it would also be of interest that new studies focus on one of the fundamental factors for communicative effectiveness, which is buying behaviour.

## Figures and Tables

**Figure 1 behavsci-12-00151-f001:**
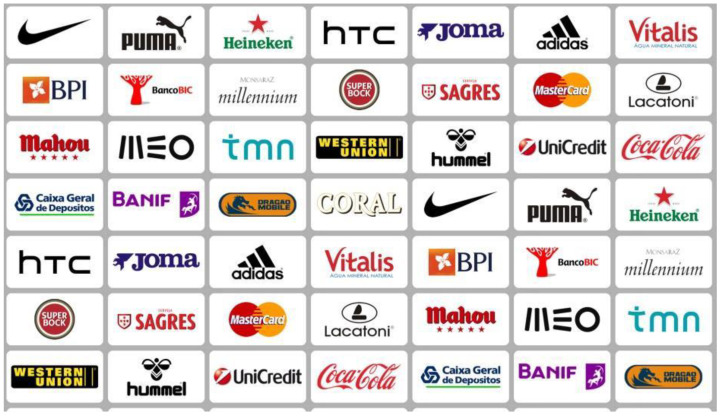
Marks placed in the experiments with up to 50 logos.

**Figure 2 behavsci-12-00151-f002:**
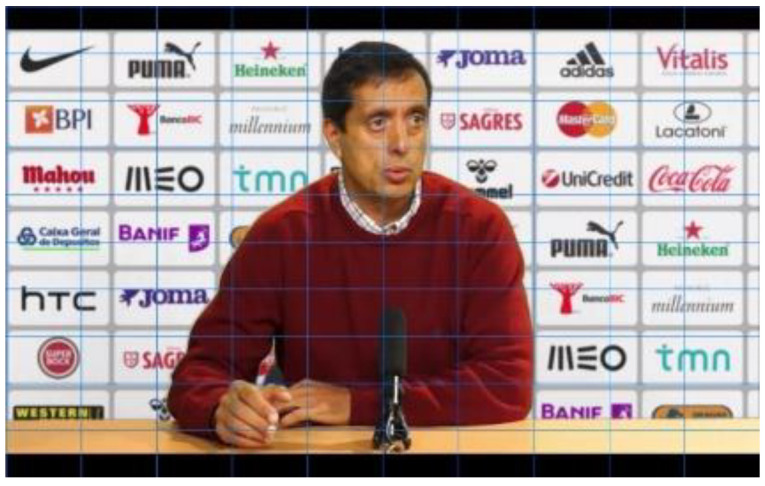
Stimulus plan with up to 50 logos.

**Figure 3 behavsci-12-00151-f003:**
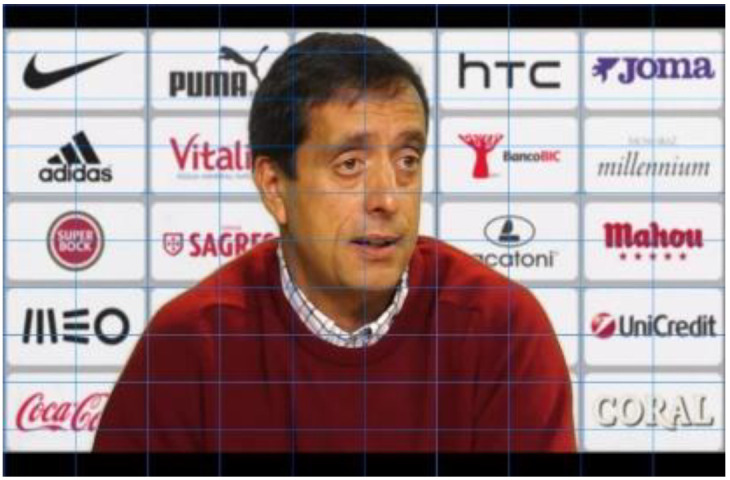
Stimulus plan with up to 25 logos.

**Figure 4 behavsci-12-00151-f004:**
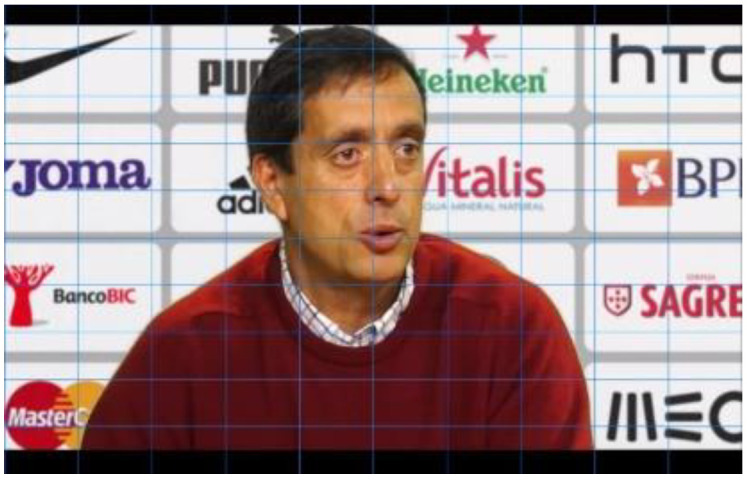
Stimulus plan with up to 12 logos.

**Table 1 behavsci-12-00151-t001:** Ranks.

	Time	N	Mean Rank	Sum of Ranks
RE	1 min	16	15.78	252.50
	2 min	16	17.22	275.50
	Total	32		
RA	1 min	16	14.16	226.50
	2 min	16	18.84	301.50
	Total	32		

**Table 2 behavsci-12-00151-t002:** Test Statistics ^a^.

	RE	RA
Mann–Whitney U	116.500	90.500
Wilcoxon W	252.500	226.500
Z	−0.446	−1.435
Asymp. Sig. (2-tailed)	0.656	0.151
Exact Sig. [2*(1-tailed Sig.)]	0.669 ^b^	0.160 ^b^
Exact Sig. (2-tailed)	0.668	0.156
Exact Sig. (1-tailed)	0.334	0.078
Point probability	0.007	0.003

a. Grouping Variable: Time, b. Not corrected for ties.

**Table 3 behavsci-12-00151-t003:** Correlations.

			RE	RA
Spearman’s rho	total_segs	Correlation Coefficient	0.702 **	0.687 **
		Sig. (2-tailed)	0.000	0.000
		N	24	24

**. Correlation is significant at the *p* = 0.001 (2-tailed).

**Table 4 behavsci-12-00151-t004:** Ranks.

	Amount	N	Mean Rank	Sum of Ranks
RE	12–25	16	12.81	205.00
	25–50	16	20.19	323.00
	Total	32		
RA	12–25	16	12.97	207.50
	25–50	16	20.03	320.50
	Total	32		

**Table 5 behavsci-12-00151-t005:** Correlations.

			RE	RA
Spearman’s rho	Amount	Correlation Coefficient	0.411 *	0.388 *
		Sig. (2-tailed)	0.019	0.028
		N	32	32

*. Correlation is significant at the *p* < 0.05 (2-tailed).

**Table 6 behavsci-12-00151-t006:** Correlations.

			RE	RA
Spearman’s rho	total_US	Correlation Coefficient	0.724 **	0.709 **
		Sig. (2-tailed)	0.000	0.000
		N	24	24

**. Correlation is significant at the *p* < 0.001 (2-tailed).

**Table 7 behavsci-12-00151-t007:** Correlations.

			RE	RA
Spearman’s rho	US_Pont_Tot	Correlation Coefficient	0.747 **	0.732 **
		Sig. (2-tailed)	0.000	0.000
		N	24	24

**. Correlation is significant at the *p* < 0.001 (2-tailed).

**Table 8 behavsci-12-00151-t008:** Ranks.

	Amount	N	Rank Mean	Sum of Ranks
Fixations	12–25	16	12.38	198.00
	25–50	16	20.63	330.00
	Total	32		

**Table 9 behavsci-12-00151-t009:** Test Statistics ^a^.

	Fixations
Mann–Whitney U	62.000
Wilcoxon W	198.000
Z	−2.491
Asymp. Sig. (2-tailed)	0.013
Exact Sig. [2*(1-tailed Sig.)]	0.012 ^b^
Exact Sig. (2-tailed)	0.012
Exact Sig. (1-tailed)	0.006
Point probability	0.000

a. Grouping variable: Amount, b. Not corrected for ties.

**Table 10 behavsci-12-00151-t010:** Correlations.

			Fixations
Spearman’s rho	Amount	Correlation Coefficient	0.447 *
		Sig. (2-tailed)	0.010
		N	32

*. Correlation is significant at *p* < 0.05 (2-tailed).

**Table 11 behavsci-12-00151-t011:** Correlations.

			Total_Fixs
Spearman’s rho	total_US	Correlation Coefficient	0.790 **
		Sig. (2-tailed)	0.000
		N	24

**. Correlation is significant at the *p* < 0.001 (2-tailed).

**Table 12 behavsci-12-00151-t012:** Correlations.

			Fix_Tot
Spearman’s rho	US_Pont_Tot	Correlation Coefficient	0.845 **
		Sig. (2-tailed)	0.000
		N	24

**. Correlation is significant at the *p* < 0.001 (2-tailed).

**Table 13 behavsci-12-00151-t013:** Correlations.

			RE	RA
Spearman’s rho	Fix_tot	Correlation Coefficient	0.572 **	0.635 **
		Sig. (2-tailed)	0.004	0.001
		N	24	24

**. Correlation is significant at the *p* < 0.001 (2-tailed).

**Table 14 behavsci-12-00151-t014:** Ranks.

	Phase	N	Rank Mean	Sum of Ranks
Mean	Exploratory	170	97.03	16,495.50
	Experimental	32	125.23	4007.50
	Total	202		

**Table 15 behavsci-12-00151-t015:** Test Statistics ^a^.

	Mean
Mann–Whitney U	1960.500
Wilcoxon W	16,495.500
Z	−2.504
Asymp. Sig. (2-tailed)	0.012

a. Grouping Variable: Phase.

**Table 16 behavsci-12-00151-t016:** Correlations.

			RE	RA
Spearman’s rho	Atittudes	Correlation Coefficient	0.676 **	0.688 **
		Sig. (2-tailed)	0.000	0.000
		N	24	24

**. Correlation is significant at the *p* < 0.001 (2-tailed).

**Table 17 behavsci-12-00151-t017:** Correlations.

			Unknown	Strange
Spearman’s rho	TOM	Correlation Coefficient	0.414 *	0.445 *
		Sig. (2-tailed)	0.044	0.029
		N	24	24

*. Correlation is significant at the *p* < 0.05 (2-tailed).

## Data Availability

Not applicable.

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
