# Peer review of "The Communicative Effectiveness of Branding at Sports Press Conferences"

_behavsci, 2022, doi:10.3390/bs12050151_

Round 1

Reviewer 1 Report

with so many hypotheses put forward, it is worth asking more than one research question that will more broadly define the scope and significance of the research conducted. Which will allow for a thorough verification of the conducted research. It is worth expanding the literature review on the subject

Author Response

Dear Reviewer,

Thank you very much for contributing to the improvement of our article. Your help has been fundamental in making this research better and with more quality.

Best Regards

Reviewer 2 Report

I have some major methodological concerns about this study. 

First of all the title misrepresents the study. This is not a study about branding at sport press conferences, but a simulated conference. The title should represent this as hopeful readers will be disheartened when they find out this is a lab experiment when they expect a real life study.

Second no details are given about the simulation regarding the use of the logo's (are they existing logo's or fictional ones, have the test subjects prior knowledge of the logo's, what about the use of color, contrast - next to size and position...?). In this respect this study cannot be independently replicated by another group to corroborate these findings. 

If real life logo's were used, it is necessary to rate (posthoc) the appreciation of the participants for the brands that are represented by the logo (positive, neutral, negative) as this may have an impact on recollection.

Third the random selection of the study participants is another difference with real life settings. In real life only people with a major interest in the sport will look at these press conferences. That type of interest will 'colour' their appreciation of the branding that is shown. There is also no control group of people who are exposed to other images (other images, other texts, reversed logo's) or who are asked to perform another mental task (rate the emotional status of the 'presenter') to compare their recollection of the used logo's. 

In the simulated scenario there is also no place for the emotional state of the person who observes the logo's (anticipation before a match, feeling of joy or sadness after a match because also there press conferences are given). If a title such as this one is used, those are factors that need to be taken into account.

In research it is sometimes necessary to work with a simulation, but on this topic that removes too many factors that play a major role in the appreciation of the branding. I would advocate a more realistic type of simulation (or the use of real press conferences with fans and controls) on this topic.

Author Response

(The authors gave the same response as above.)

Reviewer 3 Report

I appreciate researchers efforts and suggest minor changes

Abstract

Your started with methodological approach. Add aims of research first then go ahead

Introduction

Introduction must show a state-of-art and explaination of need for your research. The first 3 lines are ok. After that add the urgency of your current research. i.e. shy it is needed?

Line 64 and line 74

Don’t use words like “our” just say this study or current research.

Also add implication of study in hand?

  1. Methodology 77

2.1 Research problem and hypotheses

Follow your headings

Where is research problem

Add it

Analysis is conducted well

Extend conclusion thorugh adding more excited results and contributions of your study i.e. who can get benefits from this research ? how your extended previous work etc

Author Response

(The authors gave the same response as above.)

Round 2

Reviewer 2 Report

The authors have adequately responded to the reviewer comments.